# Frequency of Restaurant, Delivery and Takeaway Usage Is Not Related to BMI among Adults in Scotland

**DOI:** 10.3390/nu12092501

**Published:** 2020-08-19

**Authors:** Ahmad Albalawi, Catherine Hambly, John R. Speakman

**Affiliations:** 1School of Biological Sciences, University of Aberdeen, Tillydrone Ave, Aberdeen AB24 2TZ, UK; r03aaaa@abdn.ac.uk (A.A.); c.hambly@abdn.ac.uk (C.H.); 2State Key Laboratory of Molecular Developmental Biology, Institute of Genetics and Development Biology, Chinese Academy of Sciences, Beijing 100101, China; 3Centre of Excellence in Animal Evolution and Genetics, Chinese Academy of Sciences, Kunming 650223, China

**Keywords:** food outlet usage, obesity, energy intake, energy contents

## Abstract

Background: The frequency of visits to restaurants has been suggested to contribute to the pandemic of obesity. However, few studies have examined how individual use of these restaurants is related to Body Mass Index (BMI). Aim: To investigate the association between the usage of different types of food outlets and BMI among adults in Scotland. Method: The study was cross-sectional. Participants completed an online survey for seven consecutive days where all food purchased at food outlets was reported each day. We explored the relationship between BMI and usage of these food outlets. Results: The total number of participants that completed the survey was 681. The BMI of both males and females was not related to frequency of use of Full-Service Restaurants (FSRs), Fast-Food Restaurants (FFRs), delivery or takeaways, when assessed individually or combined (TFOs = total food outlets). Conclusion: These cross-sectional data do not support the widespread belief that consumption of food out of the home at fast-food and full-service restaurants, combined with that derived from deliveries and takeaways, is a major driver of obesity in Scotland.

## 1. Introduction

Obesity is a major driver of morbidity and chronic illnesses such as type 2 diabetes, cardiovascular and musculoskeletal diseases, certain cancers and adverse psychological well-being [1]. In developed countries, the combined prevalence of overweight and obesity among adults ranges from 40% to 60% in the last ten years [2]. The U.K. health survey in 2018 indicated 63% of adults aged over 18 were overweight or had obesity [3]. In Scotland, the frequency of body mass index (BMI) over 25 increased in men from 65% to 68%, and in women from 60% to 63% between 2008 and 2018 [4].

The substantial health and financial burdens that result from overweight and obesity are expected to escalate in the future [5]. Overweight and obesity result from a positive energy balance where intake exceeds expenditure [6]. Unhealthy eating behaviors and/or low levels of exercise are thought to both contribute to the pandemic of obesity [7], although the impact of diet is generally presumed to be greater [8]. In terms of dietary factors, frequent intake of foods that are high in fat, processed carbohydrates and energy dense foods are hypothesized to lead to excess weight gain [9,10].

The frequency of visits to fast-food restaurants has been suggested to contribute to obesity due to the high energy density, low micronutrient density and large portion sizes in meals from such establishments [11,12]. It has been estimated that in the USA, adults obtain 11.3% of their daily energy intake from fast-food meals [13,14], while in the U.K. adults may obtain approximately 10% of their daily energy intake from this source [15]. These studies suggest that fast-food restaurants could be risk factors contributing to increasing obesity in the population. However, this relationship could also be influenced by factors such as socioeconomic status, which may distort the association [16,17,18]. Moreover, access to healthy options in some types of restaurants could be beneficial for consumers’ health [19]. A study in the U.S. showed the prevalence of obesity had no association with the density of fast-food or full-service restaurants after adjusting for various socioeconomic factors [16]. Moreover, our previous work showed that obesity among U.K. adults, based on the U.K. Biobank data, was not associated with the local density of fast-food or full-service restaurants, apart from fish and chip shops [20].

Relatively few studies have considered the frequency of usage of diverse aspects of the retail food environment including takeaway outlets, delivery services and full-service restaurants and how such usage may be a driver of adiposity. In the present study, we aimed to investigate the association between the usage of different types of food outlets and BMI among adults living in Scotland.

## 2. Methodology

### 2.1. Study Design

This was a cross-sectional study. Ethical approval was obtained from the Ethics Review Board of the College of Life Sciences and Medicine from the University of Aberdeen (CERB/2018/08/1601). Volunteers were invited to participate in the study through social media, by encountering individuals in the main street of Aberdeen city and by distributing flyers to postal addresses in Aberdeen. Once verbal consent was obtained, they were sent a link to the initial sociodemographic survey: (Appendix A). This survey included an electronic consent statement prior to commencing the questions. There was a further statement of agreed participation before each daily food outlet use questionnaire. In this study, we included males and females who were 18 years old or above. We excluded females who were pregnant.

In the sociodemographic survey, BMI was estimated based on participants’ self-reported weight in kilograms divided by their self-reported height in meters, squared. Data on potential confounding variables were also collected. This included the participants’ postcode district, which allows an estimate of their deprivation level based on the Carstairs index. The Carstairs index is based on four factors from the U.K. census: poor social class, lack of vehicle ownership, overcrowding and male unemployment, and the general index represents the material deprivation of a region compared to the remainder of Scotland. Indices may be positive or negative, with negative results indicating a greater affluence in the region and positive scores suggesting a comparatively greater level of deprivation. Participants were asked to disclose additional demographic information including sex, age, ethnicity (White, Asian, Black or mixed), number of people living at their household, workplace and employment status (employed, unemployed or student). In this survey, we also asked the participants about their dietary habits with 6 options and one open choice (regular diet, vegetarian, vegetarian but avoid eggs, vegetarian but avoid eggs and milk, fruitarian, pescatarian and other). We asked whether there were any food allergies, as this might affect their use of food outlets, and one final question asked about their physical activity with four options: inactive, slightly active, moderately active and highly active. One question regarding pregnancy was included.

Once they submitted their responses to survey 1, which collected basic demographic data, they were sent a link to survey 2 each day over 7 consecutive days (the food outlet usage survey, Appendix A). In this survey, we asked the participants whether they used any of food outlets or services over the previous 24 h with five options of fast-food restaurant (FFR), full-service restaurant (FSR), delivery, takeaways or none. To stimulate participants’ adherence with survey 2, an auto-reminder using the TextMagic website (https://www.textmagic.com) with a link to survey 2 was automatically generated at 8.30 p.m. for 7 consecutive days. Those who wished to stop their participation, were able to send the word “NO” and they then stopped receiving further reminders.

If the participant indicated that they had visited a food outlet in the previous 24 h, they were provided with a table that included the types of the four restaurants and services and beside each type there were options of whether they were visited for breakfast, lunch, dinner or a snack.

### 2.2. Statistical Analysis

All the participant responses were anonymized and coded using Microsoft Excel in preparation for data analysis. Sociodemographic descriptive data were collected and translated into mean, standard deviation and total percentage. The sociodemographic variables are shown in Table 1. We corrected the BMI for potential confounding factors (age, sex, ethnicity, household size, employment, workplace, dietary habits, place of living, physical activity and deprivation level). The correction of the BMI was by deriving the residuals from the general linear model (GLM) and adding them back to the mean BMI.

We counted the number of meals consumed at different types of food outlets or services (FSRs, FFRs, delivery services and takeaways) and we combined the total number of meals consumed at these premises (total food outlet usage, TFOs). We explored the relationship between the unadjusted and adjusted mean BMI and the food outlet usage using analysis of variance (ANOVA). In addition, we reanalyzed our data after excluding the participants who reported that they were students to investigate whether this changed the pattern of the relationship.

We segregated the data based on sex to investigate the association between the frequent usage of food outlets and BMI within males and females. To reduce the familywise error rate due to multiple testing for ANOVA, we used the Bonferroni correction for the *p*-value. The familywise error rate was α*_fw_* = 1 − (0.95)^5^ × 100 = 22.6% and the corrected *p*-value was 0.01 to maintain the confidence in our set of analyses. SPSS version 24 (IBM Corp, Armonk, New York, NY, U.S.) was used for analysis.

## 3. Results

### 3.1. Characteristics of the Participants

One hundred and ninety-nine participants were removed due to incomplete 7 day surveys. The total number of participants who dropped out with no reason was 42, and one participant withdrew due to pregnancy. The final number of participants who completed all seven days of the survey was 681 (Figure 1).

### 3.2. Food Outlet Usage versus Sex

The descriptive data of the participants are presented in Table 1. The mean age was 25.6 ± 9.8 standard deviation (SD) years. Females represented 57.3% (391) of the study population. The mean household size was 2.9 and the proportion of houses that included people under 17 years old was 0.25. Regarding employment status, 64.2% of the participants reported that they were employed, 5.1% were unemployed and 30.6% were students (Table 1). With respect to workplace, 88.9% reported that they worked in Aberdeen, those who reported that they worked in Aberdeenshire represented 1.6% and those in flexible premises constituted 5.1% (Table 1). The percent of participants who worked from home was 4.4%. The deprivation level in the study area averaged—1.3. This equaled six out of ten on the decile scale (Carstairs index). Ethnicities in our data were divided into four categories, Asian, Black, White and mixed, where the White was dominant (72%), followed by Asian (8.8%) and mixed (15.7%), and Black was the lowest 2.2% (Table 1).

Regarding dietary habits, 81.7% reported that they followed a regular diet with no specific restrictions, while 13% were vegetarians (Table 1). Eighty-eight percent of participants had no food allergy and 11.9% indicated they had an allergy. Out of 681, 48.3% reported that they were moderately active, those who reported that they were slightly active represented 34% and 12% said they were highly active. Inactive participants only represented 5%. The mean unadjusted BMI in our study was 26.2 ± 4.16 kg/m^2^.

We explored the difference between males and females in their usage of food outlets. We found that males used FSRs significantly more than females (mean difference per week =0.21 times, T = 2.49, *p* < 0.01). Moreover, males used FFRs significantly more frequently than females (mean difference per week =0.33, T = 3.40, *p* < 0.001).

However, no significant difference between the sexes in the use of delivery and takeaways was observed (delivery: difference = 0.11, T = 1.78, *p* = 0.07; takeaways: difference = 0.01, T = 0.24, *p* = 0.80). The total food outlet usage was significantly higher among males than that among females (difference = 0.67, T = 3.67, *p* < 0.0001) (Figure 2).

### 3.3. Food Outlet Usage and Unadjusted BMI Based on Sex

Males frequency of usage of FRSs, FFRs, delivery and takeaways was not associated with unadjusted BMI (FSR: F(6,284) = 1.67, *p* = 0.12, R^2^ = 3.41%; FFR:; F(7,283) = 1.61, *p* = 0.13, R^2^ = 3.84%; delivery: F(4,286) = 0.15, *p* = 0.96, R^2^ = 0.21%; or takeaways: F(4,286) = 1.01, *p* = 0.40, R^2^ = 1.39%) (Figure 3A–D). When the frequency of usage was combined across all outlets, there was also no association (TFOs; F (8282) = 0.75, *p* = 0.64, R^2^ = 2.08%) (Figure 3E) (Appendix A).

Among females, the usage of FSRs, FFRs, delivery and takeaways was also not significantly associated with unadjusted BMI (FSR: F(5,384) = 2.34, *p* = 0.04, R^2^ = 2.9%; FFR: F(6,383) = 2.48, *p* = 0.02, R^2^ = 3.7%; delivery: F(4,385) = 1.98, *p* = 0.09, R^2^ = 2.01%; takeaways: F(5,384) = 1.17, *p* = 0.32, R^2^ = 1.50% (Figure 4A–D) or TFOs: F(9,380) = 1.03, *p* = 0.41, R^2^ = 2.2% (Figure 4E)) (Appendix A).

### 3.4. BMI versus Socioeconomic Variables (Possible Confounding Factors)

We explored several possible confounding factors that may influence the BMI of the participants (Table 2). The variables were included individually in a GLM model, and the BMI was adjusted for those that were significant. There was a significant positive association between BMI and age (β = 0.08, *p* < 0.0001, R^2^ = 4.2%); the older the subject the higher the BMI. Sex was also significantly associated with BMI; with males having higher BMI than females did (β = 0.84, *p* < 0.0001, R^2^ = 4.01%). No association was noticed between ethnicity and BMI (*p* = 0.17, R^2^ = 0.73%). There was no significant relation between the number of people per household and BMI (*p* = 0.33, R^2^ = 0.14%). However, we found the mean BMI was significantly associated with employment status, where unemployed participants showed a positive association (β = 0.96, *p* < 0.04) while students were negatively related (β = −1.1, *p* < 0.0001) and those who were employed had no association (β = 0.15, *p* = 0.5). The whole model for employment explained 2.3% of the variation in BMI.

Regarding place of work, there was no association between the mean BMI and the workplace whether working in Aberdeen city, rurally in Aberdeenshire, or in flexible premises that changed from day to day, or whether they traveled to work or worked from home (online) (Table 2). The mean BMI was slightly but significantly higher among participants who reported that they do not follow any specific diet regime (β = 1.09, *p* < 0.01), whilst those who reported that they are vegetarians had a lower mean BMI (β = −0.05 (*p* < 0.02). The explained variation in BMI by dietary habits was 1.4%.

Regarding place of living, we did not find a difference in the mean BMI between those who lived in Aberdeen city or the surrounding area (Table 2). Moreover, no significant difference in mean BMI was found among self-reported physical activity groups (inactive, slightly active, moderately active and highly active (Table 2). The mean BMI was negatively associated with the level of deprivation based on the Carstairs index (β = −0.15, *p* < 0.005, R^2^ = 1.5%).

We adjusted the BMI using stepwise regression to include all the factors mentioned previously in the model. The variation explained by the GLM model was 8.1%. The most significant factors that the BMI was adjusted for were age, sex, dietary habits and deprivation level.

### 3.5. Food Outlet Usage and Adjusted BMI Based on Sex

In males, the frequency of use of FSRs, FFRs, delivery or takeaways when assessed individually, or combined (TFOs) was not associated with increases in the adjusted BMI (FSR: F(6,284) = 1.65, *p* = 0.13, R^2^ = 3.36%; FFR: F(7,283) = 1.65, *p* = 0.16, R^2^ = 3.59%; delivery: F(4,286) = 0.24, *p* = 0.90, R^2^ = 0.19%; takeaways: F(4,286) = 0.65, *p* = 0.62, R^2^ = 0.91%; TFOs: F(8,282) = 0.85, *p* = 0.56, R^2^ = 2.2%) (Figure 5A–E) (Appendix A). The same was observed in females. There was no significant association between greater frequencies of use of FSRs, FFRs, delivery, takeaways or TFOs and the mean adjusted BMI (FSR: F(5,384) = 1.75, *p* = 0.12, R^2^ = 2.2%; FFR: F(6,383) = 2.28, *p* = 0.03, R^2^ = 3.4%; delivery: F(4,385) = 0.21, *p* = 0.90, R^2^ = 3.07%; takeaways: F(5,384) = 1.24, *p* = 0.28, R^2^ = 1.59%; TFOs: F(9,380) = 1.02, *p* = 0.42, R^2^ = 2.3%) (Figure 6A–E) (Appendix A).

### 3.6. Food Outlet Usage vs. Unadjusted and Adjusted BMI Based on Sex, Excluding Students

We reanalyzed the data after excluding the students to investigate whether this changed the pattern of the relationship between the frequency of usage of different types of food outlets and the unadjusted and adjusted BMI. We found no significant association between the frequent usage of the included food outlets and the unadjusted or adjusted BMI after excluding the students from the dataset.

## 4. Discussion

Our analysis shows that male and female usage of FSRs, FFRs, delivery, takeaways and TFOs was not associated with unadjusted BMI. After adjusting the BMI for several possible confounding socioeconomic factors, there was still no significant association. Furthermore, we reanalyzed the data after excluding students, and no associations were noticed. This reanalysis was implemented as students may have unusual consumption behaviors [21].

Our work is consistent with several previous studies. A study in the U.S. found there was no association between fast food consumed out of the home and obesity. This latter study estimated the average energy content in the top five meals purchased from FFRs and FSRs in the USA combined with the number of meals consumed by individuals during a year [16]. This showed the contribution of the energy intake at these restaurants covered 15.9% of energy requirements [16]. Another study in Brazilian urban areas showed 18% of the total energy requirements came from food purchased out of the home [22], and consistent with our study, there was no difference in body weight among the participants whether they ate more frequently inside their homes or outside [22]. Moreover, in the USA, a study of 2156 adults showed that out-of-home food consumption was not associated with BMI [23]. Additionally, living closer to restaurants was not related to body weight among 10,199 Canadian participants and did not necessarily increase their consumption at these outlets [24]. In the U.K. there was no significant association between living near fast-food restaurants and BMI among four hundred thousand participants, using Biobank data [25].

The absence of such a relationship between BMI and fast-food restaurant usage may occur for several reasons. First, since the above studies [19,24] suggest that less than 1/5th of energy intake is consumed in these establishments, the main dietary habits that drive overweight and obesity may in fact reside in what people consume for the remaining 80% of their requirements [15]. Second, one of these possible reasons is that customers who visit food outlets, may choose more healthy options than what they eat at home [19,26]. Some behavioral studies that have focused on restaurant customers have indicated that patrons often tend to select healthier menu items and enjoy visiting restaurants offering healthy options such as brown rice, vegetarian or vegan meals [19,26,27]. The absence of an association does not support the suggestion that consumption of poor food out of the home is amplified by at-home food habits [12,24,28,29].

The finding that individuals who visit these establishments more frequently do not have greater overweight or obesity is also consistent with studies that indicate the population levels of obesity are not greater in areas where there is a higher density of such establishments. This has been demonstrated by analysis at the county level across the USA [16] as well as using U.K. Biobank data at the level of postcode district in the U.K. [20].

However, other studies have suggested such associations do exist. In Cambridgeshire, an investigation found that people who live nearer to takeaways have higher intake of food by 5.3 g per day in comparison with the ones who are less exposed [30]. Although a 5.3 g increase in the total food weight might be statistically significant in a large sample, it represents only about 50 kJ of additional energy (assuming a water content of 50% and an energy density of the remainder of 20 kJ/g) which is less than 1% of daily energy requirements and unlikely to be responsible for an increase in BMI. Moreover, the 5.3 g value itself was based on an extrapolation from food-frequency questionnaires, which are extremely inaccurate [31,32,33]. It was also noted in a U.K.-based study that food consumption at restaurants, cafes and takeaways may increase daily energy intake by between 3.2% and 4.4% in adults [34]. However, this research depended on four-day food records selected from the U.K. National Diet and Nutrition Survey without reporting whether these percentages were also related to increased BMI.

Previous studies have suggested that the association between the frequency of food outlet use and obesity could be different in males and females [30,31,34,35]. Some of these studies concluded that the association between food outlets and BMI was only significant for women [36], whilst other investigations noted that the associations between fast-food restaurants and diet [37] and BMI were more observable in men [38]. Our study is consistent with a previous U.K.-based investigation that concluded that there is no indication for sex differences in the association between BMI in males and females and their use of restaurants [39].

Finally, contributing to the contradictory results in the literature are factors that influence the relationship such as age, ethnicity, household size, employment, workplace, dietary habits, place of living, physical activity and deprivation level and how these are accounted for in the analysis. It was noted in a systematic review in 2009 that, when considering the relationship between eating out and weight gain, it is important to investigate whether food intake at restaurants is causal to overweight or merely a proxy for other unhealthy lifestyle factors that may cluster, such as physical activity and neighborhood sociodemographic status [40]. In our study, the lack of an association was evident when we used unadjusted data and data adjusted for these potential confounders.

### Strengths and Limitations

A strength of our study was that we used the TextMagic website to generate automated text-survey messaging. The benefit of using such a technique is to reduce the risk of memory-dependent error. By reminding individuals every day to complete the survey, the possibility of forgetting an event (eating at a restaurant or ordering takeaway and delivery over the previous 24 h) was likely reduced compared to asking people to recall visits to restaurants over the previous week—requiring recall of events up to seven days previously.

We counted the number of meals consumed from several different types of restaurants and food services, which strengthens the investigation of the association between BMI and frequency of usage of food outlets. However, this study also has some limitations that need to be recognized. First, the weight and height were self-reported and are subject to potential bias and error [41]. However, a study conducted in Scotland to assess the validity of self-report weights and heights in the Scottish population that included 865 men and 971 women reported that the Scottish population have a low error and unbiased reporting of their weight and height, which would be satisfactory for monitoring the prevalence of overweight and obesity [42]. Second, we did not ask people exactly what they ordered or ate to make a better evaluation of their actual energy consumption. Third, because this was an observational study that depended on the natural behavior of the participants, the number of participants with extremely high restaurant usage was relatively small. Finally, we emphasize that not finding associations does not necessarily mean individuals could eat a high quantity of meals from these outlets without health consequences.

## 5. Conclusions

In this study, we evaluated the association between the frequency of use of different types of food outlets and BMI in males and females in the U.K. No association was found between FSRs, FFRs, delivery, takeaways and TFOs and BMI in both males and females. These data do not support the widespread belief that consumption of food out of the home at fast-food and full-service restaurants, combined with that derived from deliveries and takeaways, is a major driver of obesity. Our study was however cross-sectional and did not include measures of actual food consumption. Further experimental studies are needed to confirm the importance (or not) of food intake from these sources for the risk of developing obesity.

## Figures and Tables

**Figure 1 nutrients-12-02501-f001:**
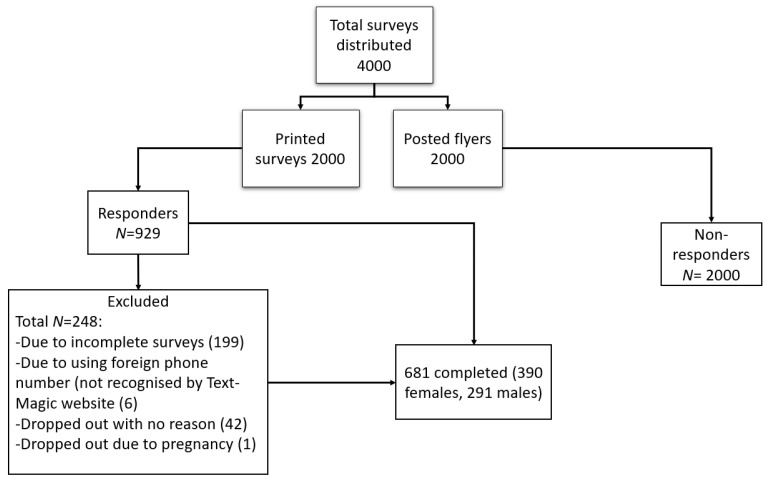
Flow chart of inclusion and exclusion of participants.

**Figure 2 nutrients-12-02501-f002:**
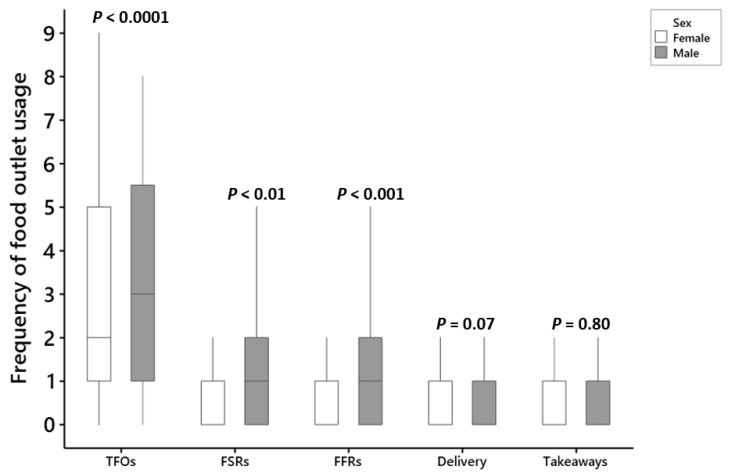
Comparison between the usage of different types of food outlets among males and females, FSRs = full-service restaurants, FFRs = fast-food restaurants, TFOs = total food outlets.

**Figure 3 nutrients-12-02501-f003:**
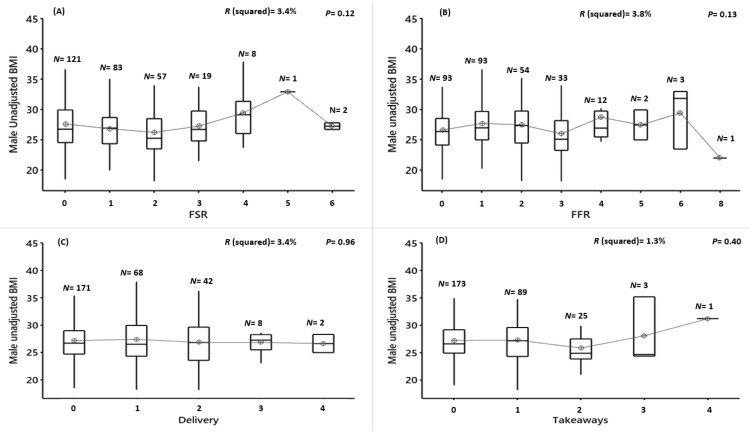
Male unadjusted BMI vs. the frequency of food outlet usage for 7-consecutive days. (**A**) FSR = full-service restaurant, (**B**) FFR = fast-food restaurant, (**C**) delivery, (**D**) take-away and (**E**) TFOs = total food outlets, *N* = number of participants in each group. Results of the ANOVA are shown. Significance is where *p* < 0.01 (after Bonferroni correction).

**Figure 4 nutrients-12-02501-f004:**
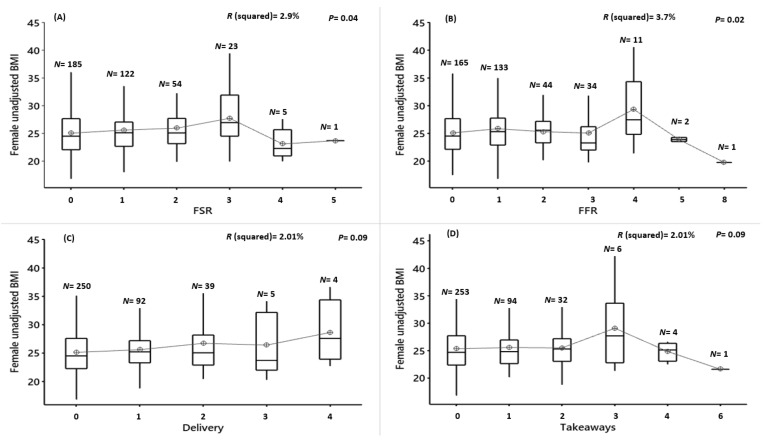
Female unadjusted BMI vs. the frequency of food outlet usage for 7-consecutive days. (**A**) FSR = full-service restaurant, (**B**) FFR = fast-food restaurant, (**C**) delivery, (**D**) take-away and (**E**) TFOs = total food outlets, *N* = number of participants in each group. Results of the ANOVA are shown. Significance is where *p* < 0.01 (after Bonferroni correction).

**Figure 5 nutrients-12-02501-f005:**
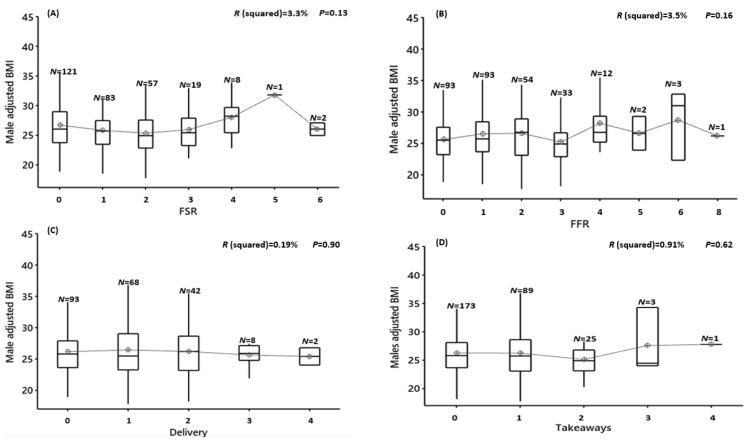
Male adjusted BMI vs. the frequency of food outlet usage for 7-consecutive days. (**A**) FSR = full-service restaurant, (**B**) FFR = fast-food restaurant, (**C**) delivery, (**D**) take-away and (**E**) TFOs = total food outlets, *N* = number of participants in each group. Results of the ANOVA are shown. Significance is where *p* < 0.01 (after Bonferroni correction).

**Figure 6 nutrients-12-02501-f006:**
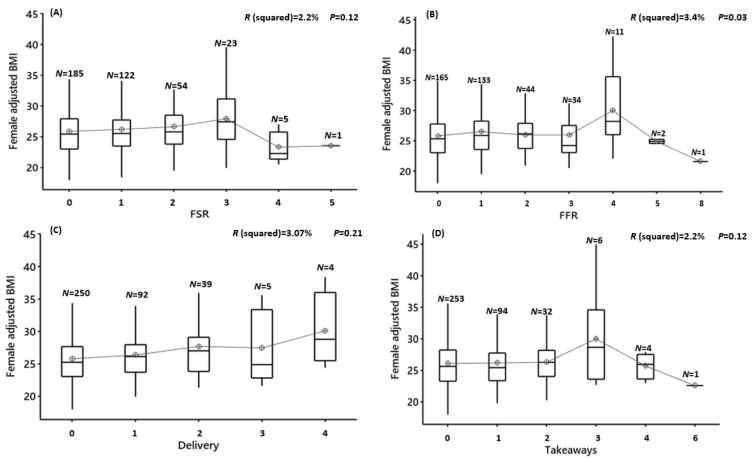
Female adjusted BMI vs. the frequency of food outlet usage for 7-consecutive days. (**A**) FSR = full-service restaurant, (**B**) FFR = fast-food restaurant, (**C**) delivery (**D**) take-away and (**E**) TFOs = total food outlets, *N* = number of participants in each group. Results of the ANOVA are shown. Significance is where *p* < 0.01 (after Bonferroni correction).

**Table 1 nutrients-12-02501-t001:** Descriptive statistics: sociodemographic characteristics of the study participants.

**Age: Mean (Standard Deviation)**	25.6 (9.8)
**Sex: number (%)**	
Females	391 (57.3)
Males	291 (42.7)
**BMI: Mean (Standard Deviation)**	
Females	25.4 (4.14)
Males	27.1 (3.99)
**Household size: Mean (Standard Deviation)**	2.9 (1.7)
**People** **under 17 in a household: Mean (Standard Deviation)**	0.25 (0.65)
**Employment: Number (%)**	
Employed	438 (64.2%)
Unemployed	35 (5.1%)
Student	209 (30.6%)
**Workplace: Number (%)**	
Work in Aberdeen	606 (88.9%)
Work in Aberdeenshire	11 (1.6%)
Work from home in Aberdeen	23 (3.4%)
Work from home in Aberdeenshire	7 (1.0%)
Flexible premises (working places change daily)	35 (5.1%)
**Deprivation level (Carstairs Index): Mean (Decile Scale)**	−1.3 (6)
**Race: Number (%)**	
White	497 (72.9%)
Asian	60 (8.8%)
Black	18 (2.6%)
Mixed	107(15.7%)
**Dietary habits: Number (%)**	
Regular diet	557 (81.7%)
Vegetarian	89 (13%)
Vegetarian but avoid eggs	1 (0.1%)
Vegetarian but avoid eggs and milk	10 (1.5%)
Fruitarian	1 (0.1%)
Pescatarian	24 (3.5%)
**Allergy**	
Yes (%)	81 (11.9%)
No (%)	599 (88%)
**Physical activity: Number (%)**	
Highly active	85 (12.5%)
Moderately active	330 (48.4%)
Slightly active	232 (34%)
Inactive	35 (5.1%)

BMI: body mass index.

**Table 2 nutrients-12-02501-t002:** General linear model analysis: BMI versus socioeconomic factors.

Factors	Β	BMI (SD)	*p*-Value	R^2^ (%)
**Age**	0.08	26.2 (4.1)	0.0001	4.26
**Sex**				4.01
Females	−0.84	25.4 (4.1)	0.0001	
Males	0.84	27.1 (3.9)	0.0001	
**Ethnicity**				0.73
White	−0.18	26.1 (4.2)	0.17	
Asian	−1.047	25.3 (4.2)	0.03	
Black	1.050	27.4 (4.3)	0.16	
Mixed	0.179	26.5 (3.7)	0.66	
**Household size**	−0.0905	26.2 (4.1)	0.33	0.14
**Employment**				2.34
Employed	0.151	26.5 (4.1)	0.594	
Unemployed	0.964	27.3 (4.7)	0.044	
Students	−1.11	25.2 (3.8)	0.0001	
**Workplace**				0.71
Work in Aberdeen City	−0.386	26.1 (4.09)	0.421	
Work in Aberdeenshire	1.18	27.7 (5.2)	0.271	
Flexible premises	−1.215	25.3 (3.3)	0.089	
Work from home in Aberdeen City	0.819	27.3 (5.3)	0.316	
Work from home in Aberdeenshire	−0.40	26.1 (6.5)	0.30	
**Dietary habits**				1.43
Avoid milk and eggs	−1.60	23.1 (2.8)	0.114	
Pescatarian	0.561	25.8 (3.7)	0.438	
Regular diet (no restrictions)	1.093	26.4 (4.1)	0.010	
Vegetarian	−0.05	25.3 (4.08)	0.02	
**Physical activity**				0.18
Inactive	0.155	26.3 (4.1)	0.777	
Slightly active	0.10	26.2 (4.5)	0.74	
Moderately Active	0.149	26.2 (4.01)	0.595	
Highly active	−0.405	25.7 (3.7)	0.302	
**Place of living**				0.03
Aberdeen City	−0.07	26.1 (4.2)	0.673	
Aberdeenshire	0.07	26.3 (4.08)	0.673	
**Deprivation level**	−0.15	26.2 (4.1)	0.005	1.15

β = coefficient, BMI = body mass index, SD = standard deviation.

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
