# Peer review of "Frequency of Restaurant, Delivery and Takeaway Usage Is Not Related to BMI among Adults in Scotland"

_nutrients, 2020, doi:10.3390/nu12092501_

Round 1

Reviewer 1 Report

Dear Authors,

Dear Editor,

I've read a new, revised version of the manuscript. By taking into account the comments of all the reviewers, the scientific quality of the manuscript has increased significantly.

Author Response

We are grateful for this positive evaluation.

Reviewer 2 Report

This manuscript has been greatly improved following rewrite; the authors are to be congratulated. I have a few remaining concerns/comments below.

Line 15 - number of participants belongs in results.

Line 17 - I don't think that 'using auto-reminder text system' belongs in this sentence.

Line 23 - should be 'in Scotland'.

Line 55 - should be 'in Scotland'.

Line 62 - the comment about the postal flyers belongs in results.

Line 94 - Inclusion/exclusion criteria should be presented earlier in the methods; suggest after paragraph 1. Inclusion criteria include 'no mental or physical illnesses' but I cannot see a question on this in your demographic questionnaire so how can you be sure?

Lines 94-98 and Figure 1 belong in results.

Figure 1 - 'Text magic software' - not mentioned in text. Add in the appropriate place with an appropriate reference for the software.

Line 104 - did you check data for normality prior to calculating mean/SD? Transform data?

Line 137 - was the mean BMI reported here adjusted or unadjusted? As you comment in the methods that you performed adjustment you need to be clear in your manuscript what you are reporting.

Results - in general there are too many figures and tables, and non-significant findings reported in text, and this section is long. If you have fully described your planned analyses in the methods, your results section would benefit from being more concise and reporting significant findings, followed by a comment on 'no other significant findings.'

Discussion, para 2 - as you did not collect information on energy intake this section is a little misleading. While you may be correct you cannot compare your findings with this. The fact that you don't have intake information is a significant limitation but your whole aim is about frequency of usage not energy intake.

Conclusion - you conducted a cross sectional study with no objective measures; both significant limitations when you want to draw such firm conclusions. Given those limitations I think that your conclusion should direct the reader to future directions to confirm your findings rather than resting on your final sentence. 

While in general the standard of English is good this manuscript would benefit from a thorough proof read to pick up spelling and grammar errors, missing or additional words, and errors in abbreviations.

Reference list needs to be checked for consistency eg journal titles abbreviated or not.

Author Response

Thank you so much for your great review.

Line 15 - number of participants belongs in results.

Response: This has now been moved to the results, see line 17.

Line 17 - I don't think that 'using auto-reminder text system' belongs in this sentence.

Response: now it is removed.

Line 23 - should be 'in Scotland'.

Response: now changed to ‘in Scotland’. See line 23.

Line 55 - should be 'in Scotland'.

Response: now changed to “in Scotland”, see line 55.

Line 62 - the comment about the postal flyers belongs in results.

Response: now this comment has been removed from the method section.

Line 94 - Inclusion/exclusion criteria should be presented earlier in the methods; suggest after paragraph 1.

Response: now moved to the methods section, see line 65 - 66.

Inclusion criteria include 'no mental or physical illnesses' but I cannot see a question on this in your demographic questionnaire so how can you be sure?

Response: This has now been removed.

Lines 94-98 and Figure 1 belong in results.

Response: now moved to the result section, see lines 117 – 120 and figure 1 in line 138.

 Figure 1 - 'Text magic software' - not mentioned in text. Add in the appropriate place with an appropriate reference for the software.

Response: now is added with a reference, see line 88.

Line 104 - did you check data for normality prior to calculating mean/SD? Transform data?

Response: yes

Line 137 - was the mean BMI reported here adjusted or unadjusted? As you comment in the methods that you performed adjustment you need to be clear in your manuscript what you are reporting.

Response: this is before adjustment.

Comment: Results - in general there are too many figures and tables, and non-significant findings reported in text, and this section is long. If you have fully described your planned analyses in the methods, your results section would benefit from being more concise and reporting significant findings, followed by a comment on 'no other significant findings.'

Response: with respect the reviewer is confusing the importance of statistical significance with biological importance. Many of the findings we made were not statistically significant. But that is the whole point. There was no significant association between BMI and restaurant usage. This is statistically not significant but biologically important to state and show.

Discussion, para 2 - as you did not collect information on energy intake this section is a little misleading. While you may be correct you cannot compare your findings with this. The fact that you don't have intake information is a significant limitation but your whole aim is about frequency of usage not energy intake.

Response: Although we did not collect information on the % of energy consumed in these establishments that doesn’t mean we cannot discuss other studies that did this. The fact other studies showed this % is relatively low is consistent with our finding that there was no link between BMI and frequency of restaurant usage.

Conclusion - you conducted a cross sectional study with no objective measures; both significant limitations when you want to draw such firm conclusions. Given those limitations I think that your conclusion should direct the reader to future directions to confirm your findings rather than resting on your final sentence. 

Response: We now included a sentence at the end of the conclusions to make this point. see line 372, and also noted in the abstract that the study was cross-sectional.

While in general the standard of English is good this manuscript would benefit from a thorough proof read to pick up spelling and grammar errors, missing or additional words, and errors in abbreviations.

Response: we now checked the ms throughout for such errors and corrected them.

Reference list needs to be checked for consistency eg journal titles abbreviated or not.

Response: now checked.

Reviewer 3 Report

The article presents a simple, straightforward, analysis of a cross-sectional survey carried out in 681 healthy adults living in the area of Aberdeen, Scotland.  The frequency of eating in various types of prepared food outlets (full service, fast food, delivery, takeaways) is reported over 7 consecutive days. The statistical analysis finds no significant correlation between frequency of eating in prepared food outlets and BMI, after adjusting for various possible confounders (age, level of socio-economic deprivation, usual dietary habits).  This work confirms various reports from other parts of the world showing little relationship between any form of eating out and BMI. Its originality rests in the sample (Scottish adults) and in the survey methods including daily reminder messages sent to the participants in order to facilitate compliance).  The work is competent, the article is clearly written.

Here are a few comments that the authors may want to use in order to improve their paper. Most of them have to do with the clarity of the text:

There are a number of unclear sentences in the manuscript, particularly in the sections modified in the revision. Here are a few examples:

Abstract, line 13: “new technology of reminding to avoid memory error” is quite unclear. What is the reminding about? How is it supposed to act on memory?

Lines 30-31: the sentence is awkward.  “Men … increased from 65% to 68%, or “females from 60% to 63%” I suppose you mean: the frequency of BMI over 25 increased in men from 65% to 68%, and in women from 60% to 63%.

Lines 44-45: Be more explicit about the “many factors”.

Line 55: Why don’t you keep to Scotland (rather than UK)?

Lines 61-62. The added text in the revision makes the text difficult to follow. I would suggest deleting “inviting them to participate and by” and, on the following line: “the 2000 distributed postal flyers yielded no participants” (this information appears in the figure).

Line 81: delete “for the exclusion criteria”.

Lines 114-115 and section 3.6: One question that does not have to do with language: Why did you reanalyze your data after excluding students? This should be justified and addressed in the discussion section.

One further limitation that you may want to address in the discussion is the decreasing number of participants as the weekly number of events increased. In some cases you had very low Ns (for example, just one woman reported 9 instances). This does not invalidate your conclusions but deserves some discussion.

Author Response

Comment: The article presents a simple, straightforward, analysis of a cross-sectional survey carried out in 681 healthy adults living in the area of Aberdeen, Scotland.  The frequency of eating in various types of prepared food outlets (full service, fast food, delivery, takeaways) is reported over 7 consecutive days. The statistical analysis finds no significant correlation between frequency of eating in prepared food outlets and BMI, after adjusting for various possible confounders (age, level of socio-economic deprivation, usual dietary habits).  This work confirms various reports from other parts of the world showing little relationship between any form of eating out and BMI. Its originality rests in the sample (Scottish adults) and in the survey methods including daily reminder messages sent to the participants in order to facilitate compliance).  The work is competent, the article is clearly written.

Response: we are grateful for this positive overview

Here are a few comments that the authors may want to use in order to improve their paper. Most of them have to do with the clarity of the text:

There are a number of unclear sentences in the manuscript, particularly in the sections modified in the revision. Here are a few examples:

Abstract, line 13: “new technology of reminding to avoid memory error” is quite unclear. What is the reminding about? How is it supposed to act on memory?

Response: we used auto-reminder text messages that were sent every day to remind the participants to fill out the survey. Hence they only had to recall restaurant usage over the previous 24h, compared to recall over the previous 7 days. This method reduces the risk of memory related error. The ‘Text-magic’ website was used for that. See line 88.

Lines 30-31: the sentence is awkward.  “Men … increased from 65% to 68%, or “females from 60% to 63%” I suppose you mean: the frequency of BMI over 25 increased in men from 65% to 68%, and in women from 60% to 63%.

Response: Now changed, see line 30-31.

Lines 44-45: Be more explicit about the “many factors”.

Response: Now changed, see line 45.

Line 55: Why don’t you keep to Scotland (rather than UK)?

Response: now changed to Scotland, see line 55.

Lines 61-62. The added text in the revision makes the text difficult to follow. I would suggest deleting “inviting them to participate and by” and, on the following line: “the 2000 distributed postal flyers yielded no participants” (this information appears in the figure).

Response: now these two sentences were removed.

Line 81: delete “for the exclusion criteria”.

Response: now deleted.

Lines 114-115 and section 3.6: One question that does not have to do with language: Why did you reanalyze your data after excluding students? This should be justified and addressed in the discussion section.

Response: We were concerned that students may have unusual consumption behaviour relative to the general population and hence wanted to eliminate the possibility that the absence of an effect was because of including students in the sample. This is now added to the discussion, see lines 289-291.

One further limitation that you may want to address in the discussion is the decreasing number of participants as the weekly number of events increased. In some cases you had very low Ns (for example, just one woman reported 9 instances). This does not invalidate your conclusions but deserves some discussion.

Response: we now mentioned this in the limitations part of the discussion 361-364.

This manuscript is a resubmission of an earlier submission. The following is a list of the peer review reports and author responses from that submission.

Round 1

Reviewer 1 Report

This interesting paper presents the results of a 7-d survey in 682 healthy young adults asking about the use of food outlets in different categories (full service restaurant, fast food, delivery, takeaways). The results are analysed in males and females separately and show that the weekly frequency of food outlet use is between 0 and 8-9 times (Figures 2-5). From these survey data, the authors propose an analysis of the contribution of such food outlets to energy intake.  Meal energy intake is estimated from average restaurant/delivery meal energy content in the UK and energy needs are evaluated from doubly-labeled water studies.  The authors report that meals from restaurants/delivery outlets represent a small proportion (14.7% in males; 19.5% in females) of estimated energy requirements.  Correlation analyses show no relationship between frequency of food outlet use and BMI, either unadjusted or adjusted for relevant parameters (sex, age, diet habits, and social deprivation).  The results are discussed in terms of the potential contribution of out-of-home eating, particularly in fast food outlets, to weight gain and overweight/obesity. 

The paper is well written, the survey is straightforward, the methods seem competently used, and the results are clearly presented. There are a number of problems, however, with the various “estimated” values and the interpretation of the data in terms of relevance for energy balance and weight control.

The survey assesses how often the participants used any form of out-of-home food outlet over one week (self-reported).  Frequency of eating-out is the only quantitative behavior actually obtained from the participants in the present study. We have no idea of their total energy intake or dietary pattern, among other relevant information. Other data used in the analyses are estimated from average values (typical restaurant meals in the UK, etc.) or derived from other studies (energy requirements). This situation generates a number of questions and potential distortions.  For instance, energy intake from the reported out-of-home food outlets is estimated from average values of similar “meals” in the UK, rather than from the actual foods ordered at each reported eating-out event.  The same “average values” are used for males and females, although it is likely that men eat more than women in these outlets as they generally do under all circumstances. The “average” value may be too high in females and too low in males. It also appears that the same “average” meal size values were used regardless of the type of meal (breakfast, lunch, dinner, or possibly snack, which usually vary in size). This source of distortion should be amended if possible, or if this is impossible, it should be acknowledged and discussed.  Likewise, for energy needs, the data are derived from previous doubly-labelled water studies of unknown relevance to the particular population of participants in this study. More specifically, although the estimated energy needs in males seem reasonable (82.7 MJ/w), the value for women seems very low (50 MJ/w) in healthy young women who report being at least moderately active for half of them and whose average BMI suggests a high frequency of overweight/obesity in the group.  Are you sure of this estimate?  Overall, there could be important distortions between the “estimated” and actual values both in terms of energy intakes and energy needs.  This should be addressed as a limitation in the discussion. 

The authors conclude that the low contribution of out-of-home food outlets to energy needs makes it unlikely that eating out significantly contributes to weight gain or overweight.  This claim is not actually supported by the presented data.  As excellently exposed in the introduction of the paper, weight gain occurs as a result of positive energy balance.  Here, we do not know whether the total energy intake exceeded estimated energy needs and whether out-of-home eating contributed to this potential excess.  The authors’ estimations suggest that the total contribution of out-of-home meals is around 15-20 % or energy needs. Beyond the reservations expressed above, it would be important to know how this contribution fits in the total diet of the participants.  Apparently, the survey allowed respondents to indicate whether the reported out-of-home meals were breakfasts, lunches, or dinners (line 92). Although the text does not explicitly include this option, the reported events could also be snacks ingested in addition to “meals”. The results section does not present any information about these meal-pattern data.  Were out-of-home “meals” ingested as part of the meal pattern (i.e. instead of home-made meals in a “substitution” paradigm) or whether they were ingested on top of home-eating events (an “addition” paradigm)? If “substituted” for home-made meals, do out-of-home meals represent more or less ingested energy?  If “added” to the home-meal pattern, do they make energy intake “excessive” relative to energy needs? As presented, the data cannot rule out that out-of-home meals could exert more or less the same type of effect as sugar-sweetened beverages (among other contributors to the energy balance equation), which represent a sizeable proportion of daily energy intake in some populations (although clearly a small fraction of the energy needs), but still add to the total daily energy intake enough to make the energy balance positive in some individuals. I realise that it may not be possible for the authors to answer these questions, but at least the claim that given their low energy input, out-of-home food outlets cannot be blamed for weight gain should be qualified.

Figure 1. The arrows do not seem to reflect what was done. The 681 participants were selected out of the 929 responders, not from the excluded 248. What were the 2000 “Posted flyers” that led to 2000 non-responders?

Figures 7 & 8: Add r² and p values

Reviewer 2 Report

This manuscript aims to investigate associations between usage of food outlets and BMI, and the contribution of energy intake from these sources to total energy expenditure in adults living in the UK. However, these analyses use self-reported height and weight data and completely theoretical estimates of food intake and energy expenditure. Findings from this study design are highly questionable, and of little practical value, given that the authors draw firm conclusions based on entirely theoretical data. 

Reviewer 3 Report

Nutrients-862196-peer-review-v1-1

I took an interest in reviewing the manuscript "Frequency of restaurant, delivery and takeaway usage 2 is not related to BMI in UK adults and supplies less 3 than 20% of estimated energy requirements" to find out how the authors have disproved the stereotype that the use of out-of-home eating outlets encourages overweight and obesity. I appreciate very much the design and execution of the 7-day study, which also includes the use of the latest distribution channel in the catering sector (SDP Service Delivery Platforms), and interesting assumptions, calculations and statistical analyses.

The weakest part of the manuscript is the introduction. It may be short, but it should present the latest research results. Meanwhile, source No. 1 comes from 1997, source No. 2 from 1990, although an updated version of this report was published in 2003 (Report of a Joint WHO/FAO Expert Consultation, WHO Technical Report Series 916).

In L. 35, the sentence 'Over the last 40 years, the prevalence of overweight and obesity has increased markedly' comes from a publication in 2008, so it refers to the period around 1965-2005. How did overweight and obesity rates increase over the next 15 years? How big are the rates in the UK and Scotland? - this should be presented.

Besides, sentences in L. 37-38 (2011 reference) and in L. 40-41 (one source from 2010) are obesity clichés and represent a very basic knowledge. In my opinion, the introduction needs improvement. In addition, it is worth referring to scientific sources concerning changes in the offer of dishes of various types of catering establishments, which are beneficial for consumers' health.

In the quoted literature in general, the share of sources published before 2010 is significant - 15 out of 41, i.e. 36.5%.

Some detailed comments of a polemical nature:

Title: I feel that the knowledge / recognition of Scotland and the UK is comparable, so maybe it is worth "promoting" Scotland in the title? (and in L. 453).

  1. 104: Why were these 2 platforms chosen - they have the largest market share? Maybe the most frequently ordered food in Scotland is different than in the whole UK?

Other comments:

Table 1.: 1/the raw “ Household with people under 17: Mean (Standard Deviation)” is unclear in terms of values = 0.25. Maybe it's better this way: ‘People under 17 in a household’ or ‘Number of persons under 17 years old in the household’; 2/ The abbreviation WEE should be explained below the table; the same note applies to the abbreviations in Table 2.

  1. 326: in brackets (Figure 7. A & Figure 8. A); L. 332 – (Figure 7. B-E & Figure 8. B-E); L. 340 – (Figure 7. F-J & Figure 8. F-J).
  2. 360-362: to confirm this, it is better to refer to the Dietary Reference Value for energy (and other nutrients) that are being developed in each country (in UK, COMA sets nutrition requirements for the the UK population as the Dietary Reference Values )
  3. 381: this sentence should be completed; after 'Another study' add 'in Brazilian urban areas'.
  4. 386-388: the sentence ‘These data do not support the widespread belief that consumption of food out of the home at fast-food and full-service restaurants, combined with that derived from deliveries and takeaways, is a major driver of obesity’ fits perfectly as the ending of 'Conclusions'. And that's why I suggest you move it there.